# Study on key parameters of buckling deformation instability and fracture of rock beams and asymmetric distribution law of stope stress

**Zhanshan Shi**[1]*, **Hanwei Zhao**[1], **Bing Qin**[2], **Bing Liang**[2], **Gang Li**[1], **Xiuru Liu**[3], **Lifeng Jia**[4]

**1** School of Mining, Liaoning Technical University, Fuxin, China, **2** School of Mechanics and Engineering, Liaoning Technical University, Fuxin, China, **3** Yankuang Energy (Ordos) Company Limited, Ordos, China, **4** School of Flight, Anyang Institute of Technology, Anyang, China

\* shizhanshan@163.com

**Data Availability Statement:** All relevant data are within the manuscript and its Supporting Information files.

## Abstract

The moving deformation of the strata and the redistribution of stope stress after mining show asymmetrical characteristics, which do not conform to the symmetrical structural characteristics of the original rock beam fracture. To further analyze the deformation of rock beams and the asymmetry law of stope pressure distribution after strata caving, the detailed process of instability and deformation of composite rock beams before failure was revealed through similar material simulation, theoretical analysis, and numerical simulation. Through similar simulation experiments, the structural characteristics of strata caving were observed. After excavation, the caving angle near the open-off cut side of the model is greater than that on the stop-mining line side. The maximum bending moment of the rock beam is located at the open-off cut side. The rock beam fracture is located on the partial open-off cut side in the middle of the rock beam. The rock beam on the open-off cut side is easy to shear slip and not easy to hinge. The rock beam in front of the advancing direction of the working face is easily hinged. Based on the structural characteristics of strata caving, considering the thickness of the composite rock beam, the two-hinged arch mechanical model for rock beam fracture is established. On this basis, the key parameters of rock beam instability and fracture such as limit load, additional horizontal stress, limit break distance, and break position are analyzed. Based on the deformation characteristics of two hinged arches, the caving structure and the asymmetric distribution mechanism of stress redistribution during the deformation of overburden in stope are explained. Finally, the deformation of rock beam and the asymmetry of stress distribution in stope are verified by numerical calculation. The results show that the concentrated stress value of the coal pillar at the open-off cut side is greater than that in front of the working face. There is a pressure relief area behind the working face, and the pressure relief area has a certain range. The range of stress concentration area, pressure relief area, and stress value tend to be stable, and only the range of the original rock stress zone expands when the working face is advanced to a certain distance. The asymmetric distribution of compaction stress in goaf is related to the buckling deformation of strata.

**Funding:** This research was funded by National Natural Science Foundation of China, grant number 52004118; Department of Education of Liaoning Province, grant number LJ2020QNL009; Henan Province Key R&D and Promotion Special Project-Science and Technology Research, grant number 222102320240; Research Program of Science and Technology at Universities of Inner Mongolia Autonomous Region, grant number 2021GG0296.

**Competing interests:** The authors have declared that no competing interests exist.

# 1 Introduction

In mining engineering, the strata movement and the distribution law of mine pressure have always been problems explored by mining workers. Academician Qian Minggao called the inside of the stope a "black box" [1]. Due to its difficulty in exploration and direct observation, there are still problems that need to be explored in depth in the law of rock strata movement [2]. Coal mining caused the fracture movement and deflection subsidence of the overlying strata [3–6]. In the current theoretical study, the roof strata are regarded as beam structures [7–10], and a series of mechanical analysis models based on beam structure fracture have been established. As the basic unit of mine pressure analysis, whether there are other structural forms of rock fracture needs to be further explored. Therefore, based on experimental and model observations, the strata fracture structure that is more in line with the actual situation is revealed, and a new mechanical model is established. It is of great significance to further study the movement and deformation of strata and the redistribution of mine pressure.

At present, there are a lot of studies on the fracture structure characteristics and stress redistribution law of stope strata. In terms of the study of the characteristics of rock fracture structure, Academician Qian Minggao et al. [11, 12] proposed the theory of masonry beam structure based on the observation of the movement trajectory of the basic roof. Academician Song Zhenqi et al. [13] proposed the theory of transfer rock beam structure based on the observation of the roof of the goaf and the characteristics of the stope mine pressure. The theory of masonry beam and transfer rock beam provides guidance for the determination of roof pressure from the perspective of structural equilibrium, and elevates the roof control from qualitative to quantitative level. Zhang et al. [14, 15] gave the overburden and surface migration characteristics corresponding to different bearing structures, and established the mechanical models of different composite bearing structures based on the deformation structure characteristics of composite beams. Based on the basic principles of mining rock mechanics, He et al. [16] proposed the "equilibrium mining" theory and the "short cantilever beam" mechanical model. Ren [17] proposed the combined strata structure of "cantilever beam and articulated rock beam" in the overlying strata of shallow buried deep working face. Based on the mechanics model of the voussoir beam structure (VBS) of the unconsolidated arch structure, Wang et al. [18] analyzed the variation of load stress exerted by the arch structure on rock strata in the unconsolidated layers. Zhao et al. [19] built the mechanical model of the roof rock beam and analyzed the structure instability process of the roof rock beam. Ma et al. [20] Ma et al. calculated the roof pressure of the mechanical model of the short-arm beam formed by roof cutting combined with mechanical analysis, and obtained the roof failure criterion. Yavuz [21] studied the relationship between surface subsidence and stress distribution in goaf. In the study of stress redistribution in stope, Chen et al. [22, 23] analyzed the stress distribution law in front of coal wall by using a statistical damage simplification model. Xie et al. [24] considered the inclination angle of the coal seam and analyzed the peak position of the abutment pressure in front of the working face based on the stress equilibrium theory of loose media. None of the above stress redistribution studies were combined with stope symmetry analysis. In the analysis of the large structures of strata movement, Xie et al. [25] proposed the macro-stress shell structure of surrounding rock stress evolution. Shi et al. [26] assumed that the rock stratum collapsed into a semicircular evolution, and analyzed the extreme value of the abutment pressure. Liu et al. [27] analyzed the stress transfer characteristics of caving rock, proposed the concept of stope static abutment pressure, and analyzed its distribution characteristics. Xie et al. [28] investigated the effect of key stratum on the mining abutment pressure of a coal seam. Wang et al. [29] compared the stress redistributions of conventional and new split-level longwall layouts.

As the basic research unit, the rock beam fracture form and deformation rule are closely related to the stress redistribution state of the whole stope. In the study of beam structure, the beam occurs bending deformation and fracture, which usually shows symmetrical. Combined with the similar material simulation experiments done in recent years, it is found that the rock collapse structure in the experiment is asymmetrical. This state is similar to the buckling deformation and failure of a circular arc arch. Therefore, this study is carried out to further reveal the deformation and failure law and stress redistribution law of strata. It has certain theoretical research value.

## 2 Experimental study on fracture deformation characteristics of stope rock beams

At present, the main research method of stope rock beam fracture structure is similar material simulation experiment. Therefore, different mine working faces were selected to carry out similar material simulation experiments. The process of rock beam moving deformation and failure is analyzed, and the common characteristics of rock beam fracture structure are obtained. To highlight the deformation characteristics of the beam and avoid the interference of other mining factors, the experimental scheme was designed with near-horizontal coal seam mining as the engineering background. The main variables in the engineering background are mining height and rock physical and mechanical parameters.

### 2.1 Engineering background of experimental scheme design

Working Condition 1: At the 22201 working face of Shaqu mine, the simulated mining height is 2 m and 2# coal seam is mined. The 2# coal seam where the working face is located is located in the middle part of Shanxi formation. The average thickness of the coal seam is 1.07 m, and the average inclination is 4˚. The structure is simple without gangue or occasionally containing 1 layer, and the working face elevation is +396 ~ +486 m. The column diagram of the coal seam where the coalface is located and the lithology description of rock strata are shown in Fig 1A.

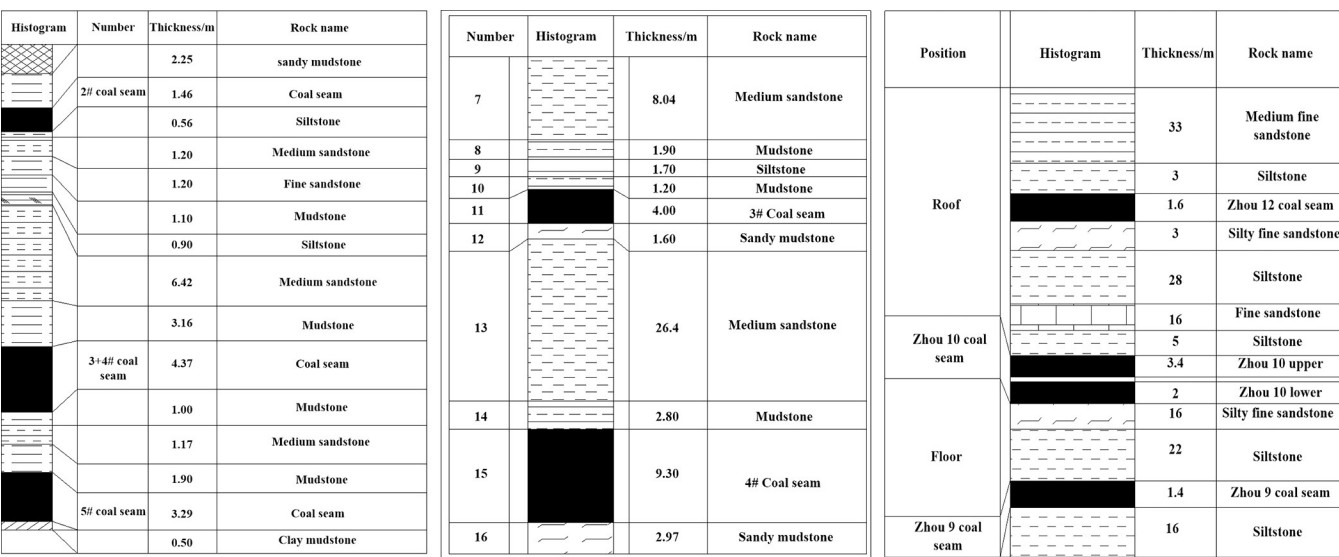

**Fig 1. Engineering background for similar simulation experiments.** (a) Column diagram of the Shaqu Mine coal seam. (b) Histogram of the coal seam at Xinan Mine. (c) Histogram of the coal seam at Daanshan Mine.

Working Condition 2: At the 1402 coalface of Xinan mine, the simulated mining height is 4 m and 4# coal seam is mined. The average burial depth of this coalface is 107 m, and the average thickness of the coal seam is 9.3 m. The inclination of the coal seam is 3~9˚, and the average is 6˚, which is a near horizontal coal seam. The column diagram of the coal seam is shown in Fig 1B.

Working condition 3: At the +400 level shaft 10 slot coalface of Daanshan mine, the simulated experimental mining height is 5 m, mining the upper and lower coal seams of shaft 10. The coal thickness on axis 10 is 0.9m~3.2 m, the average coal thickness is 3 m, and the coal seam inclination is 19˚. The average thickness under axis 10 is 2 m, and the spacing is 0.02 m~2.00 m. There is a phenomenon of local merger between the two layers, and the simulation experimental design scheme is two-layer joint mining. The comprehensive column diagram of the coal seam roof and floor is shown in Fig 1C.

## 2.2 Selection of model similarity ratios

To satisfy the requirement of experimental accuracy, the simulation similarity conditions are simplified by the main similarity factors according to the three theorems of similarity simulation. The similarity conditions between the experimental model and the prototype are mainly geometrical, kinematic similarity conditions, and dynamic similarity conditions. The size of this simulation experiment platform was 2m×0.2m×1.5m. According to the occurrence characteristics of coal and rock strata, coal seam mining height, coalface parameters, and the size of the simulation experiment platform, the size of this simulation experiment platform was 2m×0.2m×1.5m, the geometric similarity ratio of the model was determined to be 1:100. The similarity of the simulation experiment finally determined is shown in Table 1.

## 2.3 Similar simulated material ratios

According to the ratio table of similar materials in the laboratory, the materials were mainly sand, lime, gypsum, cement, mica, and other materials. The ratio numbers were selected in the ratio table in combination with the material strength to mix each component material. The above three groups of experimental ratios are shown in Tables 2 to 4, with the numbers in the ratio number being the sand, lime, and gypsum components in that order.

## 2.4 Analysis of fracture deformation law of rock beam

In this experiment, the method of photographing and sketching was used to obtain the fracture deformation law of the rock beam. The excavation step distance is 5 cm, which represents the actual working face on the site to advance 5m, and the interval between each excavation is 1.5h. Photos are taken after each excavation. Finally, the photos of the strata movement and deformation phenomenon are selected for sketching, and analyze the strata movement and deformation law.

**Table 1. Similarity ratio of the experimental model.**

| Name | Similarity ratio |
|---|---|
| Geometric similarity ratio | $C_L = L_m/L_p = 1:100$ |
| Similarity ratio of volume to weight | $C_\rho = \rho_m/\rho_p = 1:1.5$ |
| Strong similarity ratio | $C_\sigma = \sigma_m/\sigma_p = C_L \times C_P = 1:180$ |
| Time similarity ratio | $C_t = t_m/t_p = \sqrt{C_L} = 1:10$ |
| Poisson's ratio similarity ratio | $C_\mu = \mu_m/\mu_p = 1$ |

**Table 2. Similar material ratio of Shaqu mine.**

| Lithology | Rock strength/MPa | Model strength/MPa | Uniaxial Compression |
|---|---|---|---|
| | Uniaxial compression | Uniaxial compression | |
| Siltstone | 43.38 | 0.238 | 337 |
| Medium sandstone | 28.11 | 0.156 | 437 |
| Coal | 12.25 | 0.107 | 655 |
| Sandy mudstone | 32.3 | 0.190 | 455 |
| Mudstone | 26.35 | 0.156 | 537 |

**Table 3. Similar material ratio of Xinan mine.**

| Lithology | Rock strength/MPa | Model strength/MPa | Uniaxial Compression |
|---|---|---|---|
| | Uniaxial Compression | Uniaxial Compression | |
| Sandy mudstone | 15.80 | 0.09 | 473 |
| Fine-grained sandstone | 40.30 | 0.24 | 355 |
| Mudstone | 29.47 | 0.17 | 537 |
| Medium sandstone | 35.78 | 0.21 | 455 |
| Siltstone | 58.80 | 0.35 | 337 |
| Coal seam | 12.26 | 0.07 | 573 |

**Table 4. Similar material ratio of Daanshan mine.**

| Lithology | Rock strength/MPa | Model strength/MPa | Uniaxial Compression |
|---|---|---|---|
| | Uniaxial Compression | Uniaxial Compression | |
| Medium to fine sandstone | 49.65 | 0.331 | 337 |
| Siltstone | 30.44 | 0.203 | 455 |
| Shaft 12 | 9.5 | 0.063 | 673 |
| Pink fine sandstone | 55.44 | 0.370 | 337 |
| Siltstone | 62.25 | 0.415 | 337 |
| Fine sandstone | 79.16 | 0.528 | 337 |
| Siltstone | 39.75 | 0.265 | 355 |
| On axis 10 | 10.36 | 0.069 | 673 |
| Siltstone | 38.46 | 0.256 | 355 |
| Shaft 10 lower | 10.36 | 0.069 | 673 |
| Powdered fine sandstone | 55.72 | 0.371 | 337 |
| Siltstone | 31.32 | 0.209 | 455 |

**(1) Fracture deformation process analysis of rock beam at 22201 working face of Shaqu mine.** In the process of advancing the working face, the sketch of the moving damaged strata is shown in Fig 2A–2D.

The roof collapses for the first time when the working face is advanced to 65m (Fig 2A), and the caving rock stratum ranges from 0 to 2 m. The collapsed strata on the side of the working face slip and are not hinged with the roof strata, and there is a hinge phenomenon on the open-off cut side. However, the articulated structure is disconnected from the collapsed strata. The caving structure is similar to a trapezoid, trapezoid bottom side length is 60 m, height is 2 m.

The roof strata form an articulated structure when the working face is advanced to 70m (Fig 2B), and the articulated structure does not fracture obviously. It shows that the rock beam

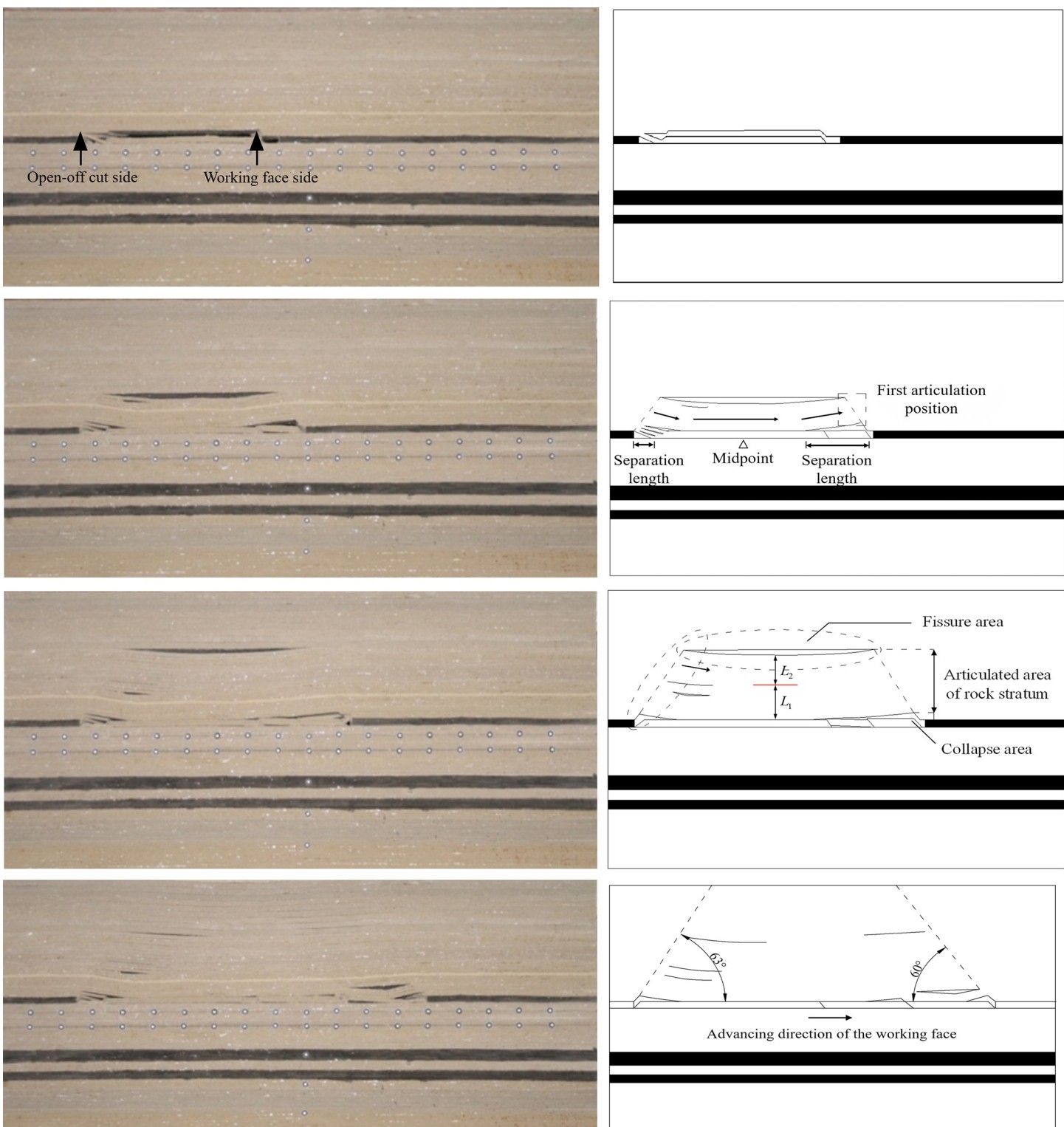

**Fig 2. Movement and deformation law of strata in condition 1.** (a) Working face advance 65m. (b) Working face advance 70m. (c) Working face advance 90m. (d) Working face advance 115m.

has a certain bending deformation ability. When the deflection does not reach the limit deflection, the rock beam is still distributed in a continuous medium state. The articulation range is 2~6.5 m above the roof, and the curved rock beam is a thick beam composed of multiple layers of rock beams. According to the sketch in Fig 2A, the midpoint of the contact area between the bending deformation of the rock beam and the underlying rock beam is inclined to the open-off cut side. The separation length formed by hinging is that the working face side is greater than the open-off cut side. The overall bending shape of the rock beam is shown by the arrow in the figure. The angle between the arrow near the open-off cut side and the horizontal direction is greater than that near the working face side.

The roof articulation structure is further extended in the strike and vertical directions when the working face is advanced to 90m (Fig 2C). The bending substructure range is extended from 2~8.5 m to 2~17.0 m. The movement range of roof strata is extended. In the figure, the rock stratum in the $L_2$ range is bent and deformed, and the curved rock beam on the working face side is bonded to $L_1$ after bending and deformation. A separation layer is formed on the open-off cut side, and the angle indicated by the arrow decreases, and the length of the separation layer increases.

The bending structure range expands to 2~33.5 m when the working face is advanced to 115m (Fig 2D). The $L_2$ region in the figure is further expanded, and no new stratification is formed subsequently. The caving angle is 63° on the open-off cut side and 60° on the working face side. The open-off cut side is slightly larger than the working face side.

**(2) Fracture deformation process analysis of rock beam at 1402 working face of Xinan mine.** The mining height of the working face is 4m. According to the sequence of strata collapse during the experiment, the pictures were marked and the caving structure as shown in Fig 3 was obtained. The numbers in the figure represent the collapse order in the strata collapse area.

When the rock beam in area 3 is fractured and deformed, it is divided into two parts. The lower part formed a separation layer on the working face side, and the open-off cut side slipped

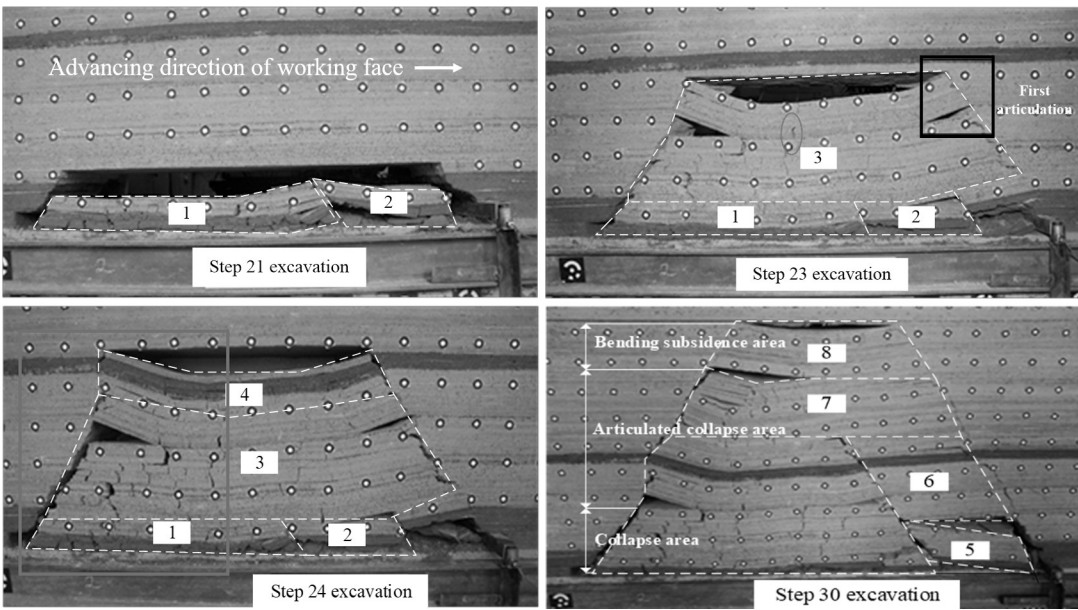

**Fig 3. Movement and deformation law of strata in condition 2.** (a)Working face advance 105m. (b) Working face advance 115m. (c) Working face advance 120m. (d) Working face advance 150m.

without forming a separation layer. The upper part formed a separation layer on both sides, similar to working condition 1. The rock beam is still asymmetrically bent, and the maximum bending moment and subsidence position are close to the open-off cut. As shown in the delineated area in Fig 3B, where the rock beam is cracked by tensile stress.

The working face is advanced to 150m, and the upper part of zone 3 continues to expand to zone 7 (Fig 3D). The newly added deformed rock strata are tightly bonded without forming an obvious separation layer. When the strata in zone 8 are fractured and deformed, a separate layer is formed on the open-off cut side. As the length of the separation layer increases, the corresponding angle between the separation layer and the horizontal direction decreases. The caving angle is 57˚ on the open-off cut side and 52˚ on the working face side. The open-off cut side is slightly larger than the working face side.

**(3) Fracture deformation process analysis of rock beam at +400 horizontal axis 10 groove working face of Daanshan mine.** The mining height of the working face is 5m. According to the sequence of strata collapse during the experiment, the pictures were marked and the caving structure as shown in Fig 4 was obtained. The numbers in the figure represent the collapse order in the strata collapse area.

The working face side is hinged when the working face is advanced to 75m (Fig 4B), and the working face side is easier to hinge than the open-off cut side. The bending deformation of the corresponding caving range 5 is still asymmetric when advancing 90m. The maximum bending moment is close to the open-off cut side, and it forms a separate layer with the strata below. With the further advance of the working face, the strata movement develops upward

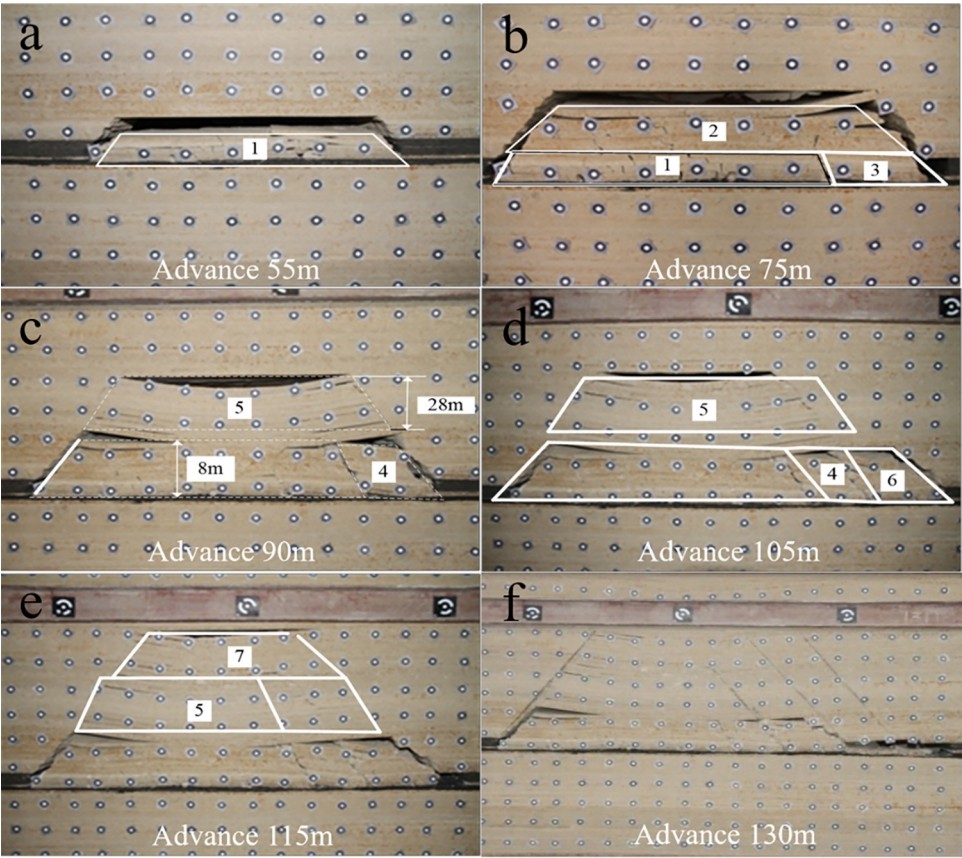

**Fig 4. Movement and deformation law of strata in condition 3.**

and no new separation layer is formed. The caving angle is 49˚ on the open-off cut side and 41˚ on the working face side. The open-off cut side is slightly larger than the working face side.

## 2.5 Summary of experimental rules

According to the above fracture and deformation laws of strata obtained by experiments with different geological conditions and different mining heights, the following general laws can be obtained: (1) After the excavation of the working face, the caving angle formed is different, and the open-off cut side is larger than the working face side. The overall caving structure is asymmetrical. (2) The multi-layer combined deformation is the main one when the rock beam is broken and deformed layer by layer, which makes it easy to form an articulated structure on the working face side. The open-off cut side is easy to slip and break, and the bending and deformation of the rock beam after forming the hinged structure is asymmetrically distributed. The maximum bending moment is located in the center of the rock beam near the open-off cut side. (3) After the articulation is formed on the working face side, with the breaking and deformation of the strata above, it is not easy to form a separation fissure on the working face side. However, the open-off cut eye is easy to form separation fissures. With the increase of the height of the caving strata formation, the Angle between the separation fissure and the horizontal direction gradually decreases. (4) After the rock beam is fractured, it can be divided into two sections, from the open-off cut to the maximum bending moment, and from the maximum bending moment to the working face side. The Angle between the first section and the horizontal direction is greater than that of the second section. (5) With the upward development of caving, the asymmetric appearance is weakened, and the separation layer on the open-off cut side is no longer produced.

## 3 Research on asymmetric distribution mechanism of rock beam fracture deformation

### 3.1 Mechanical model of rock beam fracture deformation

In the current theoretical study, the roof strata is regarded as beam structure without considering its thickness. When the rock strata are fractured and deformed with beam structure, the caving structure formed should have symmetry. Experimental research shows that when the rock beam is fractured and deformed, the structure is asymmetrical, and a new mechanical model needs to be established to explain this phenomenon. From the experimental phenomena, it can be seen that the overburden is often deformed in the form of composite rock beams when it is fractured and deformed. And the position of the maximum bending moment after deformation is close to the open-off cut side. Considering the thickness of the rock beam and the deformation characteristics of the rock beam, the two-hinged arch mechanical model as shown in Fig 5 can be established.

In the early stage of mining, when the span of the rock beam is small, with the increase of the thickness of the combined deformed strata, the upper bearing load gradually increases, and the rock beam with a certain thickness can be simplified into two hinged arches. The arch foot bears the load from above. At this time, the stress F along the axis of the arch is generated under the action of the upper load, and the stress F of the arch roof is horizontal, as shown in Fig 5A. As the span increases, the ultimate load corresponding to the two hinged arches decreases. When the ultimate load is reached, the two hinged arches become unstable and buckling deformation occurs, as shown in Fig 5B. The F-direction of the additional stress caused by the self-weight of the rock beam changes. The direction of F action is along the axis direction of the arch. The deformation of the rock beam is divided into two parts by the Z

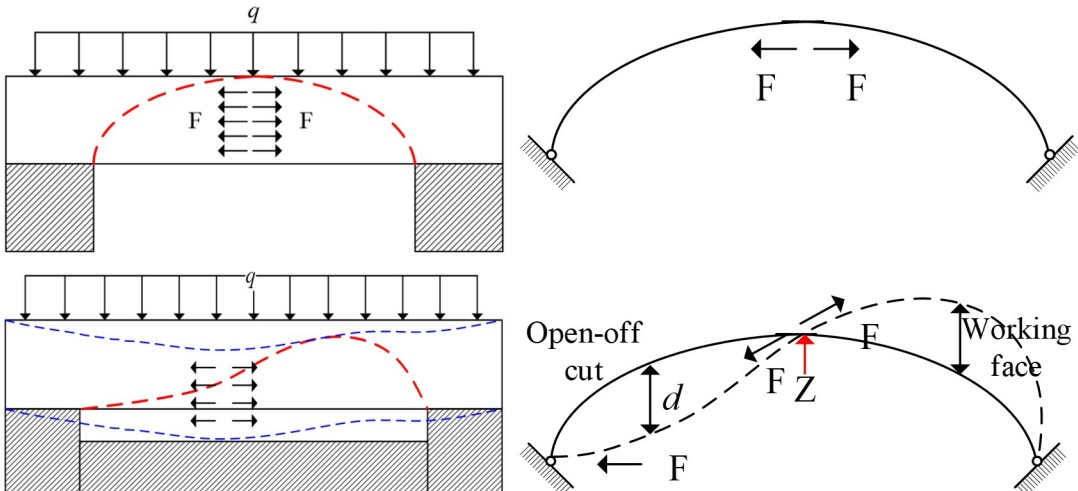

**Fig 5. Deformation process of composite rock beams.** (a) Force analysis of rock beam at the initial stage of deformation. (b) Force analysis of rock beam when it reaches bearing limit.

point. The rock beam near the cutting eye side is concave, and the rock beam near the open-off cut side is convex. Due to the restriction of the strata above, stress accumulation is easy to occur in the contact between the rock beams above.

## 3.2 Relationship between two-hinged arch structure and asymmetric deformation

**(1) The open-off cut side is concave and the working face side is convex.** The buckling deformation of two hinged arches is asymmetrical. The experimental law shows that the bending of the rock beam conforms to the buckling deformation mode shown in Fig 5B. The reason why it is concave under the open-off cut side and not on the working face side is caused by the gradual increase of load in the advancing direction of the working face during the advancing process of the working face. At the moment of buckling deformation, the corresponding load gradually increases on the rock beam along the direction of the working face. Although the load is uniformly distributed, the loading has a sequential order, which causes the buckling to be concave close to the open-off cut.

**(2) The maximum bending moment point is biased to the open-off cut side.** According to the deflection distribution after buckling and deformation in Fig 5B, there is a maximum deflection position on both sides of point Z.

**(3) The angle between the first section and the horizontal direction is greater than that of the second section.** Because the maximum deflection is biased to the open-off cut side, the fulcrum positions on both sides of the beam are the same, so the angle between the first section and the horizontal direction is greater than that of the second section.

**(4) Form an articulated structure easily on the working face side.** After buckling deformation, the flexure curve of the working face side moves upward, and the corresponding rock beam generates upward stress. Therefore, the rock beam is not easy to slip on the working face side. On the open-off cut side, the corresponding deflection curve moves downward, generating a force toward the goaf, which is easy to slip and fracture.

**(5) Not easy to form separation fissures on the working face side.** With the increase of the height range of rock collapse, the angle of the separation fracture of the open-off cut side gradually decreases. When the deformation range of the upper strata is further extended, and

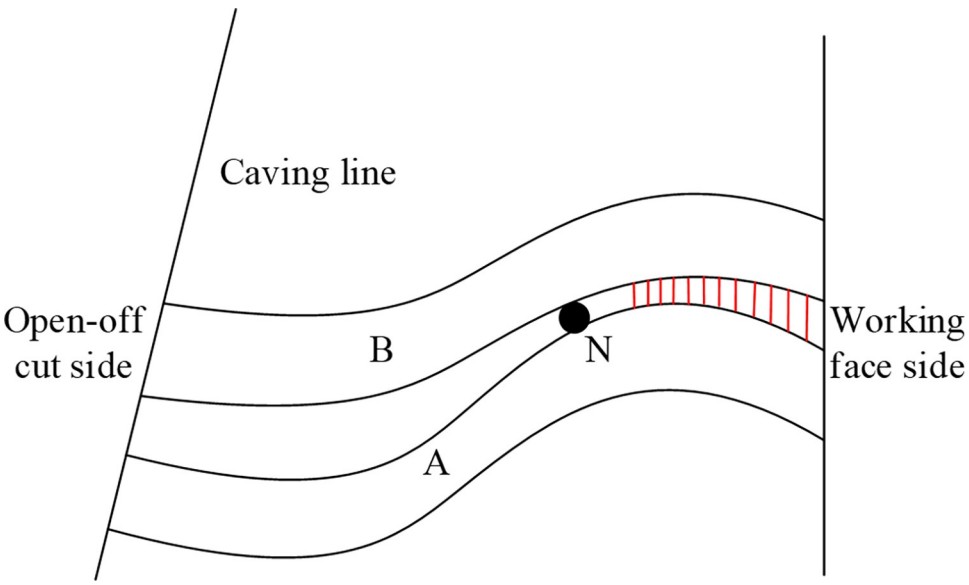

**Fig 6. Contact characteristics of adjacent composite rock beams after deformation.**

the strata strength is lower than that of the lower strata, the corresponding span is small when buckling occurs. The corresponding buckling deformation midpoint is to the left of the underlying strata. Therefore, after deformation, it can fit with the lower strata, and no obvious separation occurs in the open-off cut and the working face side, as shown in the change process of rock blocks 4 to 7 in Fig 3. When the deformation range of the upper strata expands and the strength of the extended strata is greater than that of the lower strata, the corresponding midpoint of buckling deformation moves forward than that of the lower strata, as shown in Fig 6. In this case, separation will occur on the open-off cut side. Due to the forward shift of the midpoint N, the declining space of this point is smaller than that of the underlying strata, which hinders the displacement of the extended strata. Therefore, the deflection of buckling deformation is reduced. Based on the above law, when the strata further expand and the strength is greater than the strata below, the corresponding N-point moves further forward. The length of the separation fissure on the corresponding open-off cut side will further increase. At the same time, the deflection is further reduced, resulting in a smaller angle between the separation layer and the horizontal direction. This phenomenon is consistent with the above-mentioned experimental law. Because the working face side is in the convex state and is restricted by the above rock strata, when the strata deformation continues to expand, the working face side continues to be in the compression state. The red area occurs squeeze, and it is not easy to produce a separation layer.

(6) **The caving angle of the near coal seam roof strata on the open-off cut side is greater than that on the working face side.** The author has carried out a study showing that the development of caving angle is closely related to horizontal stress [30]. The horizontal stress determines the fracture position of the strata under the load action from above. As the horizontal stress increases, the corresponding caving angle increases. According to the buckling deformation structure, when the buckling deformation of two hinged arches occurs, the corresponding beam changes, as shown in Fig 7. After deformation, the beam can be regarded as a two-stage structure, the open-off cut section is approximately a beam structure, and the working face side can be regarded as an arch structure. The additional horizontal stress values generated are equal, and most of the additional stress values on the open-off cut side are converted

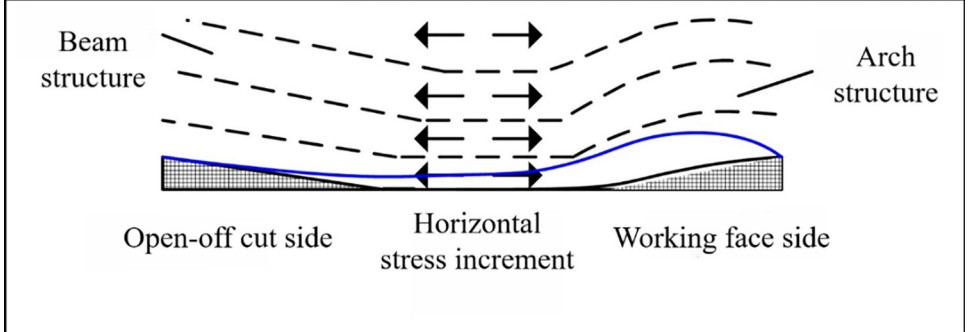

**Fig 7. Horizontal stress redistribution of rock beam after buckling deformation.**

into horizontal stresses. However, the additional horizontal stress on the working face side causes the bending deformation of the rock beam to be converted into elastic potential energy storage, and the horizontal force on the strata above the working face is small. The horizontal stress of the open-off cut side is greater than that of the working face side, resulting in a large caving angle of the open-off cut side. With the upward expansion of rock beam fracture, due to the forward movement of point N, the deformable space of rock beams decreases when there are multiple groups of rock beams above. The additional stress generated is gradually reduced, and the horizontal additional stress corresponding to the open-off cut side and the working face side is similar. The caving angles on the corresponding strata tend to be the same.

### 3.3 Key parameters of rock beam fracture deformation

**(1) Grouping of rock beams.** The rock beam is regarded as a beam with thickness, and the deformation is explained by using two hinged arches. First of all, it is necessary to determine the boundary of the thick beam, that is, to determine the grouping of the roof rock beam. The synergistic deformation characteristics of the composite rock beam are fully considered when the key strata theory is established. Therefore, when carrying out the grouping of rock beams, the key strata theory can be used for grouping.

**(2) Calculation of axial force after buckling deformation of rock beam with thickness.** The additional stresses resulting from buckling deformation can be obtained by calculating the axial force. Combined with the self-weight of the rock beam and the mechanical parameters of the rock beam, it can be determined whether the rock beam is cut off near the open-off cut side. Combined with the dilatability, the deformable space below the rock beam can be obtained, and the distribution range of the caving zone can be obtained.

**(3) Determination of bending moment of rock beam with thickness.** The bending moment distribution of the rock beam is obtained by calculating the bending moment. For the working face side, due to the restriction of the rock layer above, the bending deformation is hindered, resulting in energy accumulation. The accumulated elastic energy can be calculated based on the bending moment. For the open-off cut side, the fracture position of the rock beam can be obtained according to the bending moment, and then the distribution of vertical cracks on the open-off cut side can be determined, as shown in the blue box in Fig 3C. After the buckling deformation occurs, the maximum bending moment point corresponding to the expansion of the buckling deformation strata above is pushed forward as the working face is advanced. The corresponding vertical fracture area gradually increases and moves forward, but the whole is concentrated on the open-off cut.

**(4) Determine deflection of rock beam with thickness.** Through the deflection calculation, the deflection curve can be obtained, and the deformation form of the rock beam can be further obtained. The N point is found to determine the position of the horizontal separation layer on the open-off cut side, and the subsidence space reduction of the rock beam due to the forward movement of the N point.

**(5) Stope energy changes caused by buckling deformation.** Because the rock stratum in the caving zone is broken and scattered in the goaf, the strata in the caving zone do not accumulate energy. At the same time, the open-off cut side can effectively release the flexural elastic energy generated by the buckling deformation of the roof, and no energy accumulation occurs. Energy accumulation occurs on the working face side of the fracture zone and bending subsidence zone. The upward bending deformation of this area is hindered by the upward bending deformation and the energy accumulates. It is easy to cut off along the caving line when accumulated to a certain value, and the energy is suddenly released, causing an impact load. For inclination with the same characteristics, the ability accumulation of inclination will cause impact deformation of adjacent roadways.

## 4 Theoretical calculation method of key parameters of rock beam fracture deformation

4.1 Critical advancing distance of rock beam deformation

The initial symmetric bending deformation has little effect on the critical antisymmetric buckling load of the arch. Assuming that the arch axis is consistent with the pressure line, the branch point buckling theory is adopted. The circular arch under radial load can also represent the non-circular arch under vertical load in engineering practice, if the ratio of vector to span is not large. The instability of a curved rock beam is regarded as a circular arch problem under radial load.

As shown in Fig 8, the thickness of the curved rock beam is W, and the arch height is the strata thickness.

The differential equation for buckling in the circular arch plane expressed by the vertical displacement v is:

$$\frac{d^2 v}{ds^2} + \frac{v}{R^2} = -\frac{M}{EI_x} \tag{1}$$

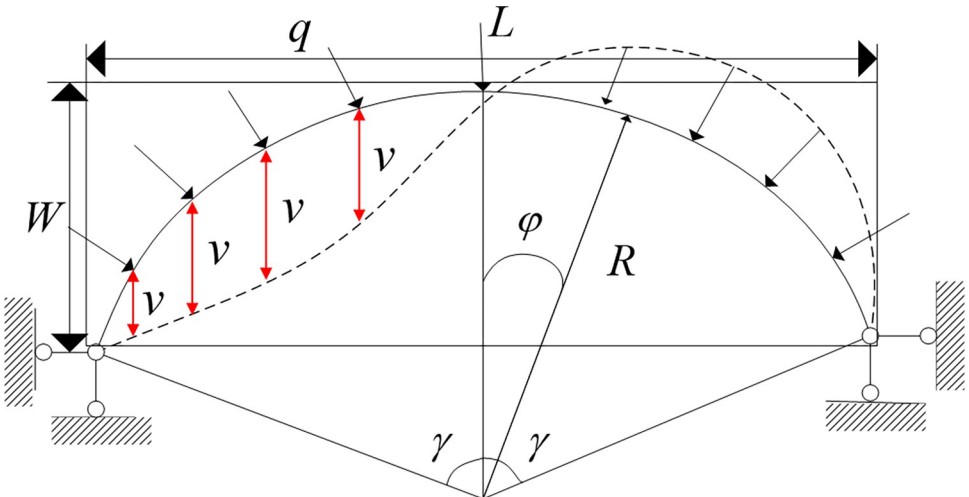

**Fig 8. Mechanical model of roof rock beam stability.**

The bending moment equation is:

$$M = -\frac{EI}{R^2}\left[C_2 + C_3(1 - \beta^2)\beta\cos\beta\varphi - C_6(1 - \beta^2)\beta\sin\beta\varphi\right] \tag{2}$$

Where $M$ is the bending moment, N·m, $EI$ is the bending stiffness, GPa; and $R$ is the arch radius; $\beta = \sqrt{1 + \frac{qR^3}{EI}}$; $C_1$, $C_2$, $C_3$, and $C_4$ are undetermined coefficients, they are given by the boundary conditions; $\varphi$ is the included angle, °.

Obtained by the singular function condition $M$ is $\varphi$,

$$M = -\frac{EI}{R^2}C_6(1 - \beta^2)\beta\sin\beta\varphi \tag{3}$$

Boundary conditions: $\varphi = \pm\gamma, M = 0$

$\sin\beta\gamma = 0$ is obtained using the boundary condition,

$$\beta\gamma = n\pi \tag{4}$$

According to $\beta = \sqrt{1 + \frac{qR^3}{EI}}$, $q_{cr} = \frac{EI}{R^3}\left(\frac{n^2\pi^2}{\gamma^2} - 1\right)$ obtaining the arch critical pressure

$$q_{cr\min} = \frac{EI}{R^3}\left(\frac{\pi^2}{\gamma^2} - 1\right) \tag{5}$$

When the rock beam thickness is $W$ and the overhanging length of the rock beam is $L$,

$$(R - W)^2 + \left(\frac{L}{2}\right)^2 = R^2 \tag{6}$$

Solution,

$$R = \frac{W}{2} - \frac{L^2}{8W} \tag{7}$$

Bringing Eq (6) into Eq (4) yields,

$$q_{cr\min} = \frac{EI}{\left(\frac{W}{2} - \frac{L^2}{8W}\right)^3} \times \left(\frac{\pi^2}{\gamma^2} - 1\right) \tag{8}$$

During the increase of $L$, the rock beam force $q$ remains unchanged. The rock beam occurs buckling deformation when $q_{cr\min} > q$, the bending rock beam is unstable. At this time, the corresponding working face advancement distance is the critical advancement distance.

## 4.2 Additional horizontal stress after deformation of rock beam

The bending subsidence strata occur buckling deformation when the working face is advanced to a certain distance. After deformation, the action of rock strata and the axial force of the arch foot are converted into horizontal stress and applied to the front of the working face. Therefore, the axial force of the bending strata can be regarded as the horizontal additional stress value when it reaches the buckling deformation.

The axial force of the arch section of a double-hinged circular arch under the action of radial load $q$.

$$N_{cr} = q_{cr} \cdot R \tag{9}$$

According to Eq (7), the critical advancing distance is $L$ when the bending deformation

occurs in the curved strata. The R-value can be calculated according to Eq (7). Bring $R$ into Eq (8) to calculate the additional horizontal stress value.

## 4.3 Calculation of maximum bending moment and position of rock beam after deformation

Take the derivative of Eq (1) with respect to $\varphi$,

$$\frac{\mathrm{d}M}{\mathrm{d}\varphi} = 0, \text{ then there is } -\frac{EI}{R^2}C_3(1-\beta^2)\beta^2\cos\beta\varphi = 0 \tag{10}$$

among $-\frac{EI}{R^2}C_3(1-\beta^2)\beta^2 \neq 0$, only $\cos\beta\varphi = 0$, $\beta\varphi = \pm\frac{\pi}{2} + n\pi$,

$$\varphi = \pm\frac{\pi}{2}\frac{1}{\beta} = \pm\frac{\pi}{2}\frac{E^2I^2}{\sqrt{E^2I^2 + qR^3}}$$

$$M_{\max} = -\frac{EI}{R^2}C_3(1-\beta^2)\beta\sin\left(\beta \cdot \pm\frac{\pi}{2}\frac{1}{\beta}\right) = \pm\frac{EI}{R^2}C_3(1-\beta^2)\beta \tag{11}$$

## 4.4 Calculation of maximum deflection of rock beam after deformation

The vertical displacement equation is:

$$v = C_1 + C_2\cos\varphi - C_3\beta\sin\beta\varphi - C_4\sin\varphi + C_5\beta\cos\beta\varphi \tag{12}$$

$v$ is an odd function condition of $\varphi$, can be obtained after simplification

$$v = C_1 - C_3\beta\sin\beta\varphi - C_4\sin\varphi \tag{13}$$

The vertical displacement at the extreme value of the bending moment is maximum, ie

$$v_{\max} = C_1 \pm C_3\beta - C_4\sin\left(\pm\frac{\pi}{2}\frac{E^2I^2}{\sqrt{E^2I^2 + qR^3}}\right) \tag{14}$$

# 5 Asymmetric distribution of stope stress based on buckling deformation structure characteristics

## 5.1 Numerical model establishment

In order to verify the non-uniform distribution of stope stress caused by buckling deformation of rock beams described in the above theoretical analysis, a numerical simulation of the working face stress distribution of Shaqu mine 24201 was carried out based on the above rock beam deformation model. The asymmetric redistribution of stope stress is further explained.

The curved rock beam is a continuous medium, so the numerical calculation software uses FlAC$^{3D}$ finite element software for simulation. At the same time, to obtain the general law, the parameters of the material model are divided into three parts, namely roof, floor, and coal seam. The material parameters of the model are shown in Table 5. To make the roof strata

**Table 5. Mechanical parameters of the model.**

| Lithology | Bulk density /(kN·m$^{-1}$) | Elastic modulus/MPa | Poisson's ratio | Internal friction angle /(˚) | Cohesion /MPa | Tensile strength/MPa |
|---|---|---|---|---|---|---|
| Roof | 26.5 | 15600 | 0.1 | 45 | 12.7 | 1.35 |
| Floor | 26.5 | 10600 | 0.13 | 40 | 13.5 | 1.26 |
| Coal seam | 26.4 | 2970 | 0.38 | 28 | 5.2 | 0.64 |

show the characteristics of bending deformation, a contact surface is set on the roof and floor of the coal seam. It enables the roof strata to be completely deformed and in contact with the floor during the excavation process.

## 5.2 Analysis of calculation results

Two working conditions were selected for simulation, namely 2 m mining height and 4m mining height. The strata mechanical parameters under different working conditions are assigned the same, and the stress contour diagram obtained is shown in Figs 9 and 10. The vertical stress distribution of the working face during the process of advancing from 0 m to 90 m is captured. The stress redistribution rule is stable when advancing to 90 m, so the subsequent cloud images are no longer listed.

In Fig 9A–9E, when the working face is advanced by 10m, the stress of the coal pillars in front of and behind the working face is concentrated, and the phenomenon of asymmetric distribution occurs. The stress concentration value behind the working face is greater than that in front of the working face. As the working face continues to advance, the stress distribution in the goaf is divided into four parts. They are stress concentration zone A in front of the working face, pressure relief zone B behind the working face, stress recovery zone C, and coal pillar stress concentration zone D of the open-off cut, as shown in Fig 9D. The range of the other partitions remains unchanged when the stress zone of the working face is stable, but the range of the stress recovery zone increases. The stress value of each zone is D>A>C>B.

In Fig 10, the mining height is increased from 2 m to 4 m, and the zonal characteristics of the overlying strata on the floor compaction stress in the mining area still exist. However, due

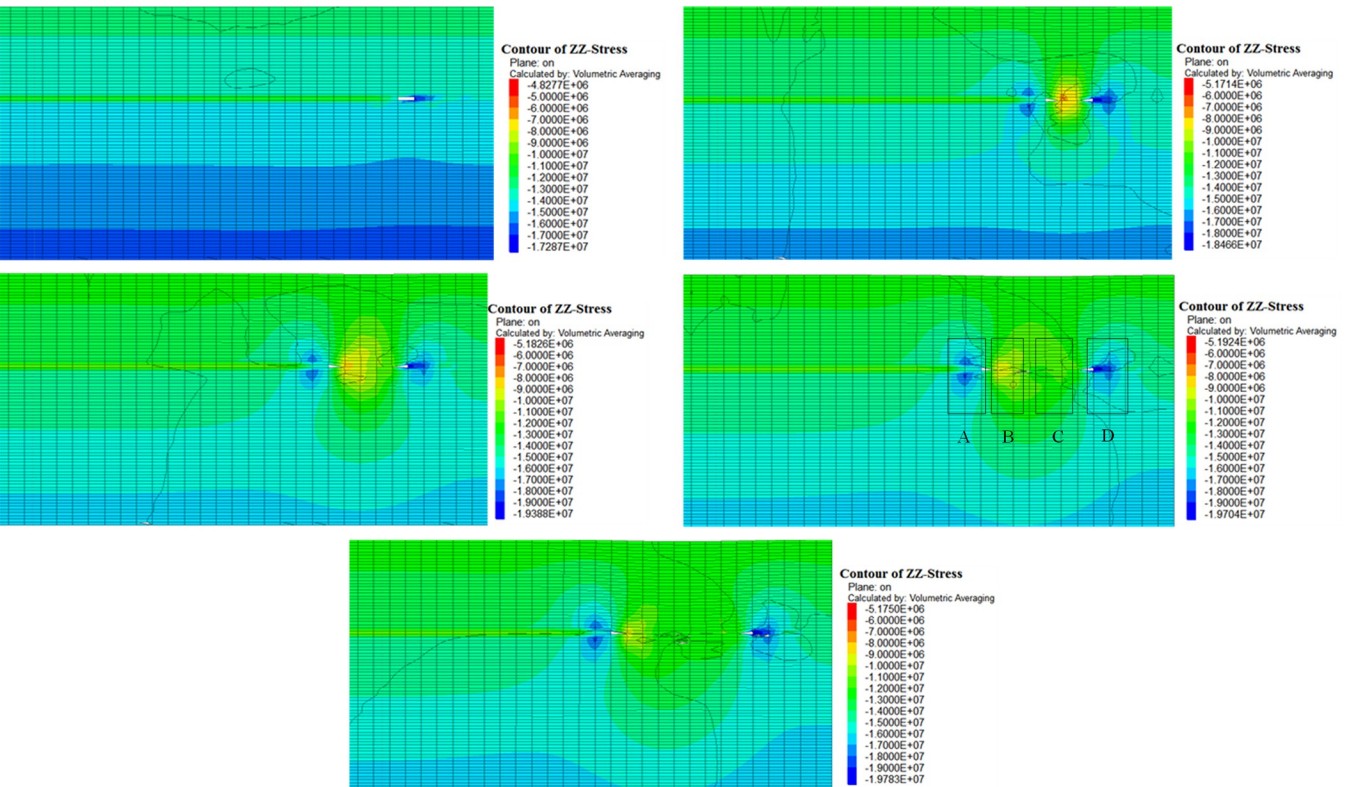

**Fig 9. Evolution of vertical stress at 2m mining height.** (a)Advance 10m. (b)Advance 30m. (c) Advance 50m. (d) Advance 70m. (e) Advance 90m.

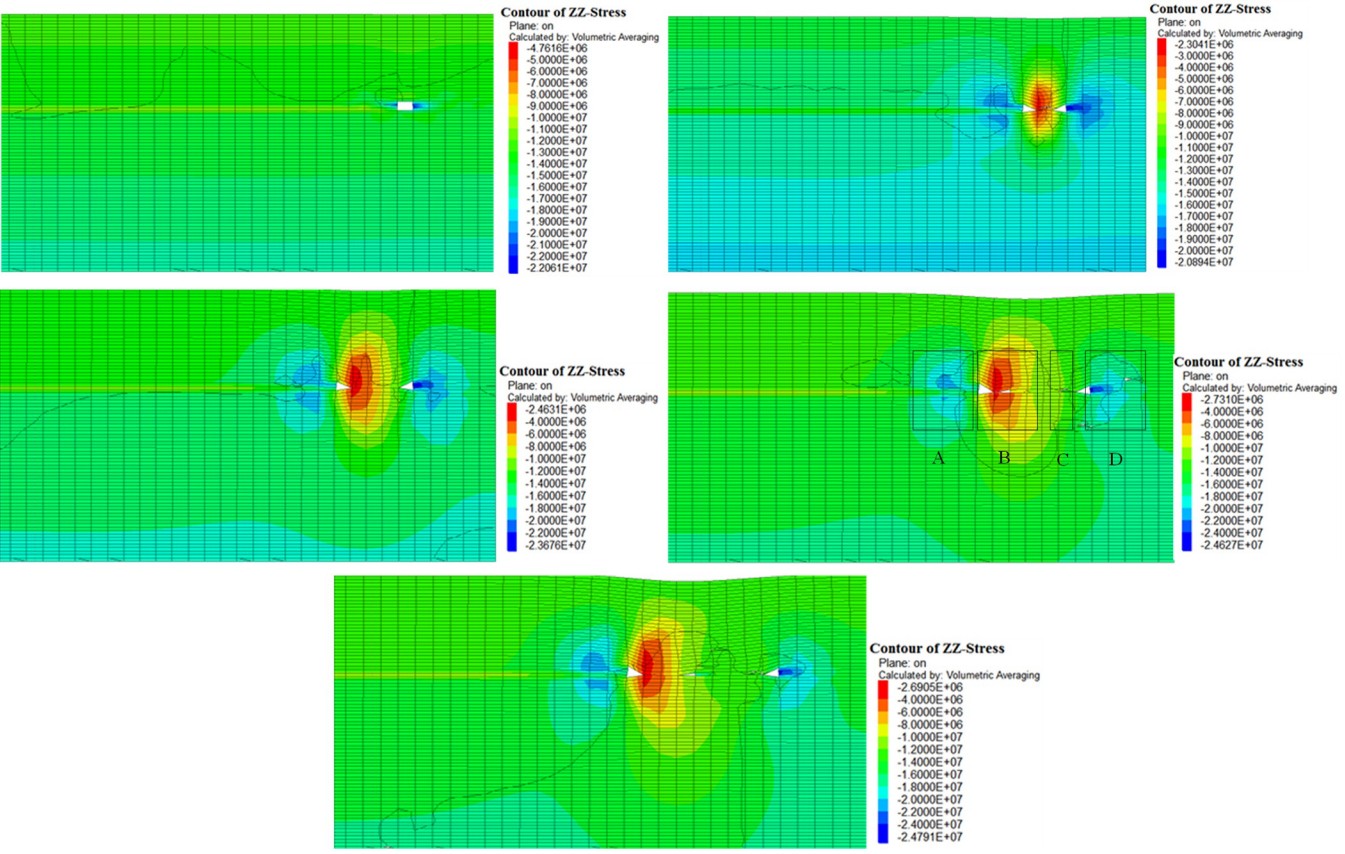

**Fig 10. Evolution of vertical stress at 4m mining height.** (a) Advance 10m. (b) Advance 30m. (c) Advance 50m. (d) Advance 70m. (e) Advance 90m.

to the increase in mining height, the vertical stress in zone A and D at 4 m mining height increases compared with that at 2 m mining height, and the vertical stress in zone B decreases compared with that at 2 m mining height. Larger mining height is conducive to the bending deformation of the strata, and the increase of deformation leads to the increase of horizontal stress near the buckling deformation. As a result, the concentrated stress of the open-off cut and the coal pillar on the side of the working face is higher than that of the 2 m mining height when the mining height is 4 m. At the same time, after buckling deformation, the goaf on the open-off cut side of the surrounding rock is in a pressure-relief state. Due to the large spatial range of buckling deformation, the upper part further expanding and deforming of rock formations range is large, resulting in the increase of the pressure relief value in the pressure relief area.

In Fig 11, the stress measurement points are evenly arranged 1 m below the bottom plate, and the horizontal spacing of each measurement point is 10 m. The values of the stress measurement points in Fig 11 also reflect the law shown in the contour diagram. During the advancing process of the working face, the horizontal stress of buckling deformation increases at first and then gradually becomes stable. The range of the upper rock strata where buckling deformation occurs gradually decreases due to the reduction of the deformation space. Therefore, for the stress concentration value in front of the working face (zone A), the area is stable after gradually increasing, as shown by the curve in the blue box in Fig 11. For zone B, the corresponding pressure relief value decreases gradually, as shown in the curve in the yellow box in Fig 11.

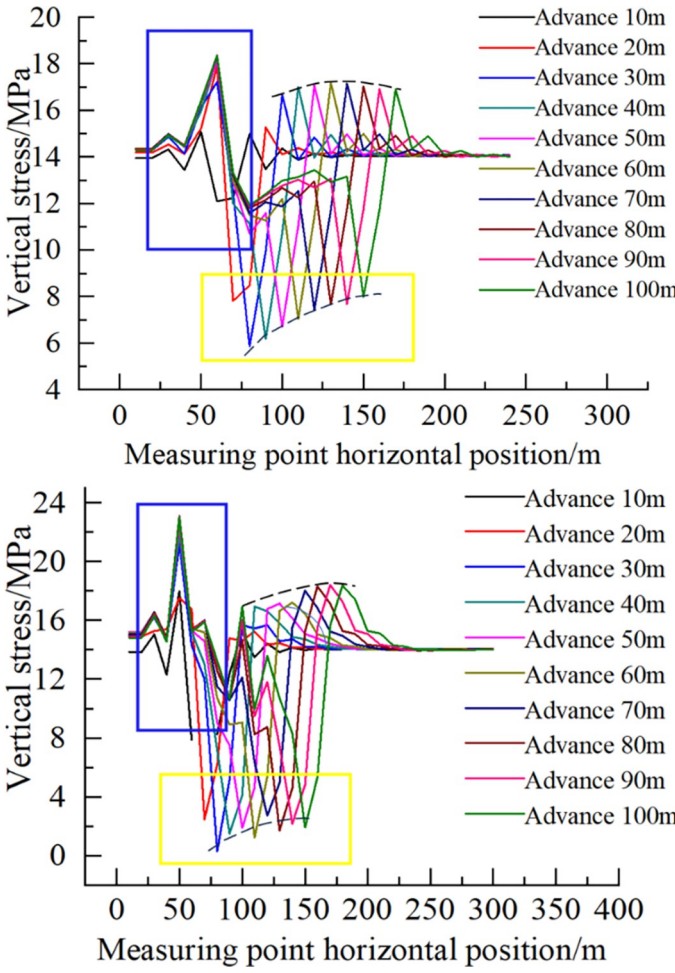

**Fig 11. Vertical stress distribution of floor at different mining heights.** (a) Mining height 2 m. (b) Mining height 4 m.

## 6 Conclusion

1. Based on the observation of strata caving structure in multiple groups of similar material simulation experiments, it is revealed that the stope strata show asymmetrical distribution structure characteristics after fracture. The caving Angle formed after mining is different, and the open-off cut side is larger than the working face side. The rock beam is easy to form a hinged structure on the working face side, and the open-off cut side is easy to slip and break. The maximum bending moment is located in the center of the rock beam near the open-off cut, and the fracture position of the rock beam is also at this position. It is not easy to form the separation fissure on the side of the working face, but it is easy to form the separation fissure on the side of the open-off cut. With the increase of the height of the caving strata group, the angle between the separation fracture and the horizontal direction gradually decreases. As the upward development of caving, the asymmetric appearance is weakened, and the separation layer on the open-off cut side is no longer produced.

2. Based on the asymmetric deformation law of rock beam fracture obtained by experiment, the mechanical analysis model of two-hinged arch structure of rock beam fracture is

established. The instability and buckling deformation characteristics of the two hinged arches are used to reveal the internal mechanism of the asymmetrical structural characteristics of the stope overburden fracture deformation. The key parameters for the study of rock beam fracture law based on the characteristics of two-hinged arch instability structure are proposed.

3. A theoretical calculation method for the key parameters of rock beam fracture deformation based on the buckling deformation of the two-hinged arch is given. It includes the critical advance distance of the rock beam after deformation, the additional horizontal stress generated by the deformation of the rock beam, the maximum bending moment and position, and the maximum deflection.

4. A numerical calculation model of bending deformation of overlying rock beams when the working face mining is established. The asymmetric redistribution of stope stress and the influence of mining height on stress distribution are given. After the bending deformation of the rock beam, the stress value of the coal pillar at the side of the open-off cut is greater than that in front of the working face, showing an asymmetric distribution law. After the increase of mining height, the concentrated stress value of the coal pillar on the open-off cut side and in front of the working increases and the pressure relief value increases. The evolution law of concentrated stress of coal pillar and pressure relief value in pressure relief area is related to the evolution law of horizontal stress caused by buckling deformation of rock beam. The influence range of rock beam buckling deformation overburden gradually expanded, and the horizontal stress first increased and then gradually stabilized. The extent of the upper rock strata where buckling deformation occurs gradually decreases due to the decrease of the deformation space. As a result, the increase of concentrated stress shows a trend of first increasing and then stabilizing with the advancement of the working face, and the pressure relief value gradually decreases with the advancement of the working face.

## Supporting information

**S1 File. Data of Fig 11.**
(XLSX)

## Acknowledgments

Many thanks to everyone involved during the experiment. The author would also like to thank Dr. Fang for his valuable comments and suggestions for improvement of the manuscript.

## Author Contributions

**Conceptualization:** Zhanshan Shi.

**Data curation:** Hanwei Zhao.

**Funding acquisition:** Zhanshan Shi, Lifeng Jia.

**Methodology:** Hanwei Zhao.

**Resources:** Bing Liang.

**Supervision:** Xiuru Liu, Lifeng Jia.

**Validation:** Gang Li.

**Visualization:** Bing Qin.

**Writing – original draft:** Zhanshan Shi.

**Writing – review & editing:** Zhanshan Shi, Hanwei Zhao.

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
