## [Decision Letter · Decision Letter 0]

24 Mar 2024

PONE-D-23-41833Study on the key parameters of buckling deformation instability and fracture of rock beams and asymmetric distribution law of stope stressPLOS ONE

Dear Dr. Shi,

Thank you for submitting your manuscript to PLOS ONE. After careful consideration, we feel that it has merit but does not fully meet PLOS ONE’s publication criteria as it currently stands. Therefore, we invite you to submit a revised version of the manuscript that addresses the points raised during the review process.

We look forward to receiving your revised manuscript.

Kind regards,

Khalil Abdelrazek Khalil, Ph.D.

Academic Editor

PLOS ONE

 [This research was funded by National Natural Science Foundation of China, grant number 52004118; Department of Education of Liaoning Province, grant number LJ2020QNL009.].  

5. PLOS requires an ORCID iD for the corresponding author in Editorial Manager on papers submitted after December 6th, 2016. Please ensure that you have an ORCID iD and that it is validated in Editorial Manager. To do this, go to ‘Update my Information’ (in the upper left-hand corner of the main menu), and click on the Fetch/Validate link next to the ORCID field. This will take you to the ORCID site and allow you to create a new iD or authenticate a pre-existing iD in Editorial Manager. Please see the following video for instructions on linking an ORCID iD to your Editorial Manager account: " ext-link-type="uri" xlink:type="simple">https://www.youtube.com/watch?v=_xcclfuvtxQ".

6. We note that Figure(s) 2, 3 and 4 in your submission contain copyrighted images. All PLOS content is published under the Creative Commons Attribution License (CC BY 4.0), which means that the manuscript, images, and Supporting Information files will be freely available online, and any third party is permitted to access, download, copy, distribute, and use these materials in any way, even commercially, with proper attribution. For more information, see our copyright guidelines: http://journals.plos.org/plosone/s/licenses-and-copyright.

1. You may seek permission from the original copyright holder of Figure(s) 2, 3 and 4 to publish the content specifically under the CC BY 4.0 license. 

Reviewers' comments:

Reviewer's Responses to Questions

**Comments to the Author**

1. Is the manuscript technically sound, and do the data support the conclusions?

Reviewer #1: Yes

Reviewer #2: Yes

2. Has the statistical analysis been performed appropriately and rigorously? 

Reviewer #1: Yes

Reviewer #2: Yes

3. Have the authors made all data underlying the findings in their manuscript fully available?

Reviewer #1: Yes

Reviewer #2: Yes

4. Is the manuscript presented in an intelligible fashion and written in standard English?

Reviewer #1: Yes

Reviewer #2: Yes

5. Review Comments to the Author

Reviewer #1: The detailed process of instability and deformation of composite rock beams before failure was revealed through similar material simulation, theoretical analysis, and numerical simulation. Based on the structural characteristics of strata caving, considering the thickness of the composite rock beam, the two-hinged arch mechanical model for rock beam fracture is established. Based on the deformation characteristics of two hinged arches, the caving structure and the asymmetric distribution mechanism of stress redistribution during the deformation of overburden in stope are explained. The calculation system of mine pressure in scientific mining is constructed based on the rock beam fracture theory. The paper is innovative and has great significance in guiding similar engineering problems. The manuscript can be accepted after minor revision.

1. Please simplify the title of Table 1. It is proposed that the title of Table 2 be changed to Similar material ratio of Shaqu Mine. Please make the same changes to Table 3 and Table 4.

2. It is suggested that the title of section 2.2 be changed to simulate experimental conditions more in line with the topic. The reason for the selection of the experimental model similarity ratio is not sufficient, not only based on the size of the experimental bench.

3. Please adjust the font of the caving angle of the sketch in Figure 2d for a clearer and more aesthetically pleasing presentation.

4. The language and structure of the paper can be improved by further editing for clarity and coherence.

Reviewer #2: In the manuscript titled “Study on the key parameters of buckling deformation instability and fracture of rock beams and asymmetric distribution law of stope stress”, Shi Zhanshan et al. analyzed the deformation of rock beams and the asymmetry law of stope pressure distribution after strata caving. The two-hinged arch mechanical model for rock beam fracture was established. It is of great significance to further study the movement and deformation of strata and the redistribution of mine pressure. The calculation system of mine pressure in scientific mining was constructed. The deformation of rock beam and the asymmetry of stress distribution in stope were verified by numerical calculation. It has certain guiding value for practical engineering.

This paper has clear logic, standard experimental analysis, and great innovation in research results. The manuscript can be accepted after minor revision.

1. What should be specified as the basis for determining the model similarity ratio, not just the basis for determining the geometric similarity ratio.

2. It is recommended to unify the font sizes of all images in Figure 2 to improve the aesthetics of the paper.

3. In Section 3.3, the expression of “thick rock beam” in titles 3 and 4 is not accurate, which is inconsistent with the above.

4. It is recommended to increase the number of references to supplement the basis of argumentation.

5. There are some language problems in the paper, which need to be polished and modified by native English speakers.

6. PLOS authors have the option to publish the peer review history of their article (what does this mean?). If published, this will include your full peer review and any attached files.

Reviewer #1: No

Reviewer #2: No

---

## [Author Response · Author response to Decision Letter 0]

13 May 2024

Reviewer 1:

1. Please simplify the title of Table 1. It is proposed that the title of Table 2 be changed to Similar material ratio of Shaqu Mine. Please make the same changes to Table 3 and Table 4.

Response 1: We think this is an excellent suggestion. According to the reviewer's comments, we have revised the title of Tables 1, 2, 3, and 4. The revised contents are on lines 145, 152, 153, and 154 of the revised manuscript.

2. It is suggested that the title of section 2.2 be changed to simulate experimental conditions more in line with the topic. The reason for the selection of the experimental model similarity ratio is not sufficient, not only based on the size of the experimental bench.

Response 2: We agree with the reviewer’s assessment. According to the reviewer's comments, we have made changes in the manuscript. The basis of similarity ratio selection is explained. The revised contents are on lines 134-142 of the revised manuscript.

3. Please adjust the font of the caving angle of the sketch in Figure 2d for a clearer and more aesthetically pleasing presentation.

Response 3: Thanks for your suggestion. According to the reviewer's comments, we have changed the font in Figure 2d to make it to convey the message more clearly.

4. The language and structure of the paper can be improved by further editing for clarity and coherence.

Response 4: Thanks for your constructive suggestion. We have carefully scrutinized the manuscript and made some corresponding revisions including grammatical errors and long sentences.

Reviewer 2:

1. What should be specified as the basis for determining the model similarity ratio, not just the basis for determining the geometric similarity ratio.

Response 1: Thanks for your suggestion. According to your nice suggestions, we have made changes. The basis of the model similarity ratio selection is explained. The revised contents are on lines 135-142 of the revised manuscript.

2. It is recommended to unify the font sizes of all images in Figure 2 to improve the aesthetics of the paper.

Response 2: We agree with the reviewer’s assessment. We have made a uniform modification of the font size of Figure 2 to make it more consistent with the requirements.

3. In Section 3.3, the expression of “thick rock beam” in titles 3 and 4 is not accurate, which is inconsistent with the above.

Response 3: Thanks for your suggestion. According to the reviewer's comments, we have made changes in the manuscript. The revised contents are on lines 377 and 388 of the revised manuscript.

4. It is recommended to increase the number of references to supplement the basis of argumentation.

Response 4: We feel great thanks for your professional review work on our article. We have added more references to make the argument more comprehensive. The newly added references are labeled 5, 6, 9, 15, and 23.

5. There are some language problems in the paper, which need to be polished and modified by native English speakers.

Response 5: Thanks for your suggestion. We have invited a friend who is a native English speaker from the USA to help polish our article. And we hope the revised manuscript could be acceptable to you.

---

## [Decision Letter · Decision Letter 1]

28 May 2024

Study on key parameters of buckling deformation instability and fracture of rock beams and asymmetric distribution law of stope stress

PONE-D-23-41833R1

Dear Dr. Shi,

We’re pleased to inform you that your manuscript has been judged scientifically suitable for publication and will be formally accepted for publication once it meets all outstanding technical requirements.

Kind regards,

Khalil Abdelrazek Khalil, Ph.D.

Academic Editor

PLOS ONE

Additional Editor Comments (optional):

Reviewers' comments:

Reviewer's Responses to Questions

**Comments to the Author**

1. If the authors have adequately addressed your comments raised in a previous round of review and you feel that this manuscript is now acceptable for publication, you may indicate that here to bypass the “Comments to the Author” section, enter your conflict of interest statement in the “Confidential to Editor” section, and submit your "Accept" recommendation.

Reviewer #1: All comments have been addressed

Reviewer #2: All comments have been addressed

2. Is the manuscript technically sound, and do the data support the conclusions?

Reviewer #1: Yes

Reviewer #2: Yes

3. Has the statistical analysis been performed appropriately and rigorously? 

Reviewer #1: Yes

Reviewer #2: Yes

4. Have the authors made all data underlying the findings in their manuscript fully available?

Reviewer #1: Yes

Reviewer #2: Yes

5. Is the manuscript presented in an intelligible fashion and written in standard English?

Reviewer #1: Yes

Reviewer #2: Yes

6. Review Comments to the Author

Reviewer #1: The author has carefully revised the manuscript. The quality of the manuscript has been improved. I don't have any more comments.

Reviewer #2: After a period of revision, this manuscript has reached the level that can be published in this journal, and it is recommended to be published.

7. PLOS authors have the option to publish the peer review history of their article (what does this mean?). If published, this will include your full peer review and any attached files.

Reviewer #1: No

Reviewer #2: No

---

## [Editor Report · Acceptance letter]

31 May 2024

PONE-D-23-41833R1 

PLOS ONE

Dear Dr. Shi, 

I'm pleased to inform you that your manuscript has been deemed suitable for publication in PLOS ONE. Congratulations! Your manuscript is now being handed over to our production team.

Kind regards, 

on behalf of

Dr. Khalil Abdelrazek Khalil 

Academic Editor

PLOS ONE